# Fluorescent protein tags for human tropomyosin isoform comparison

Will Scott[1], Vitaliia Polutranko[1], Jakub Milczarek[1], Ian Hands-Portman[2] and Mohan K. Balasubramanian[1,*]

## ABSTRACT

Tropomyosin is an important actin cytoskeletal protein underpinning processes such as muscle contraction, cell shape and cell division. Defects in tropomyosin function can lead to diseases, including some myopathies and allergies. In cells, tropomyosin molecules form coiled-coil dimers, which then polymerise end-to-end with other dimers for actin association. Tropomyosin is challenging to tag for *in vivo* fluorescence microscopy without perturbing its polymerisation interfaces. We recently developed a fluorescent tag comprising a 40-amino acid flexible linker capable of detecting tropomyosin in *S. pombe* actin cables and the actomyosin ring, and in patch-like structures that were previously unappreciated. We also used this strategy successfully to tag human TPM2.2, a prominent human muscle isoform. Here, we expanded this tool to visualise eight other human tropomyosin isoforms, using mNeonGreen, mCherry, mStayGold(E138D) and mScarlet3-H tags. All showed typical tropomyosin fluorescence, no signs of cytotoxicity and are compatible with super-resolution microscopy. These tools singly or in combination should aid detailed mechanistic investigations of tropomyosin isoforms.

KEY WORDS: Tropomyosin, Fluorescence microscopy, StayGold, mNeonGreen, mCherry, Flexible linker

## INTRODUCTION

Tropomyosin is a key F-actin binding protein, most well-known for its role as a regulator of actin–myosin contraction in muscle sarcomeres (Phillips et al., 1986; Von Der Ecken et al., 2015; Gunning et al., 2008). Tropomyosins are also present in nearly all metazoan non-muscle cells, although the exact roles they play in each case remains unknown (Gunning et al., 2008). Its function in actin architecture determination gives it importance in actin-related processes, such as in cell shape, movement and division, embryogenesis, wound healing, and in the immune system (Schevzov et al., 2005; Lees et al., 2013; Eppinga et al., 2006; McKeown et al., 2014; Nakamura et al., 1995). In muscle, tropomyosin blocks myosin binding sites on F-actin, until the cell receives an action potential and the sarcoplasmic reticulum releases calcium ions that trigger the protein troponin to shift tropomyosin, exposing myosin binding sites and initiating contraction (Lehman et al., 2020; Galinska et al., 2010). In non-muscle cells, tropomyosin regulates F-actin interactions with the actin filament severing protein cofilin as well as myosin (Ono and Ono, 2002). Tropomyosin is a right-handed α-helix, that forms left-handed coiled coil dimers with other tropomyosin monomers through ionic salt bridges and hydrophobic interactions. Dimers polymerise tail-to-head with other tropomyosin dimers into a chain that uses alanine-rich flexible regions to wrap itself around the major grooves of F-actin, interacting electrostatically with charged surface residues (Meiring et al., 2018; Hodges et al., 1973; McLachlan and Stewart, 1975; Brown et al., 2001; Lehman et al., 2020). Polymers form via hydrophobic interactions between an antiparallel four-helix bundle made of the 22 C-terminal residues of one dimer overlapping with the 11 N-terminal residues of another (Hitchcock-DeGregori and Barua, 2017; Hitchcock-DeGregori, 2008).

In humans, there are four tropomyosin genes, TPM1-4, which encode tens of different isoforms via alternative exon splicing. There are two classes of tropomyosin isoform: high molecular weight, comprised of 284 residues across eight exons, and low molecular weight, comprised of 248 residues across seven exons (Gooding and Smith, 2008; Geeves et al., 2015; Hitchcock-DeGregori, 2008). The full extent of isoform-specific function is yet to be characterised, but much is known (Gunning et al., 2015), for example TPM2.2 has been identified as a prominent muscle isoform (Jin et al., 2016), and in non-muscle cells specific isoforms have been linked to different actin cytoskeleton structures. For instance, TPM1.8 is associated with lamellipodia (Brayford et al., 2016), while TPM1.7 has been linked to filopodia and stress fibres (Creed et al., 2011; Tojkander et al., 2011). With exceptions such as the short TPM1.8 isoform (Janco et al., 2016), larger tropomyosins generally bind actin more stably than shorter isoforms (Hitchcock-DeGregori, 2008). Larger isoforms are downregulated in cancer cells, resulting in more metastatic behaviour as smaller tropomyosins favour the quicker actin remodelling needed during cell crawling (Wang et al., 2019). As well as cancer, tropomyosin has been implicated in other disorders. For example, tropomyosin mutants are linked with nemaline myopathy (Donner et al., 2002). Tropomyosin is considered a pan-allergen as the immune target underpinning a number of autoimmune disorders and allergies, including dust mite and cockroach allergies (Mor et al., 2002; Geng et al., 1998; Reese et al., 1999).

The wide-ranging importance of tropomyosin has made it a common subject of cytoskeletal research. Fluorescence microscopy of cells is a standard approach in protein studies, but there is evidence that fluorescently tagging tropomyosin for live cell imaging perturbs its behaviour. In *S. pombe*, through immunostaining of fixed cells, tropomyosin is known to be present in yeast actin patches, but common live cell tags have not detected it there (Hatano et al., 2022). We recently published the

[1]Centre for Mechanochemical Cell Biology and Division of Biomedical Sciences, Warwick Medical School, University of Warwick, Coventry CV4 7AL, UK. [2]School of Life Sciences, University of Warwick, Coventry CV4 7AL, UK.

*Author for correspondence (M.K.Balasubramanian@warwick.ac.uk)

W.S., 0009-0004-8774-2935; I.H., 0000-0002-8895-2342; M.K.B., 0000-0002-1292-8602

first tool capable of labelling tropomyosin in the actin patches of live *S. pombe*, which made use of a 40-amino acid flexible linker to fuse tropomyosin to the bright green fluorescent protein, mNeonGreen (mNG) (Hatano et al., 2022). Flexible polypeptide linkers are designed with a neutral disordered secondary structure typically comprised of smaller uncharged amino acids, such as serine and glycine, although others also used include proline and glutamine (Chen et al., 2013). Linkers are widely used to reduce steric hindrance in fusions, but rarely at significant chain length. Linker length and composition can be tuned to the needs of the application; longer linkers allow greater flexibility but are bulkier, with increased risk of unwanted entanglement. Reduced glycine content has been observed to increase linker stiffness (Van Rosmalen et al., 2017). We found N-terminally tagging *S. pombe* tropomyosin with a 40-amino acid-linked mNG allows reduced interference with tropomyosin polymerisation and localisation, and we adapted the approach to visualise the human isoform TPM2.2 (Hatano et al., 2022). Here, we build upon this work and generate tools to visualise eight further human tropomyosin isoforms, with four different fluorescent proteins (mNG, mCherry, mStayGold(E138D) and mScarlet3-H).

## RESULTS

To develop a range of fluorescent tropomyosin tags, we considered excitation and emission wavelengths and photostability, and settled on a commonly used green fluorescent protein, mNG (Shaner et al., 2013); a recently discovered photostable green fluorescent protein, mStayGold(E138D) (Ivorra-Molla et al., 2024; Hirano et al., 2022); a commonly used red fluorescent protein, mCherry (Shaner et al., 2004); and a brand new red fluorescent protein with improved-photostability, mScarlet3-H (Xiong et al., 2025).

### mNeonGreen tagging of eight human tropomyosin isoforms with a 40-amino acid linker

Previously, localisation of the mNG-40AA-TPM2.2 fusion was validated through observation of co-fluorescence with phalloidin-rhodamine-stained actin structures in mammalian cells (Hatano et al., 2022). Presently, eight further well-studied human tropomyosin isoforms were selected (Table 1) and mNG-tagged versions of each were generated and transfected into hTERT-RPE1 cells, an immortalised retinal pigment epithelial cell line (Fig. 1A). Transfectants were fixed and phalloidin-rhodamine stained for spinning-disk microscopy (Fig. 2A). All isoforms showed high levels of colocalisation with phalloidin-stained actin structures, including stress fibres, the cell cortex and focal adhesions (Fig. 2B), suggesting that the tagged tropomyosins are capable of dimerisation, polymerisation and binding actin. All isoform images give positive Pearson's correlation coefficients (r) (Fig. 2C), indicating

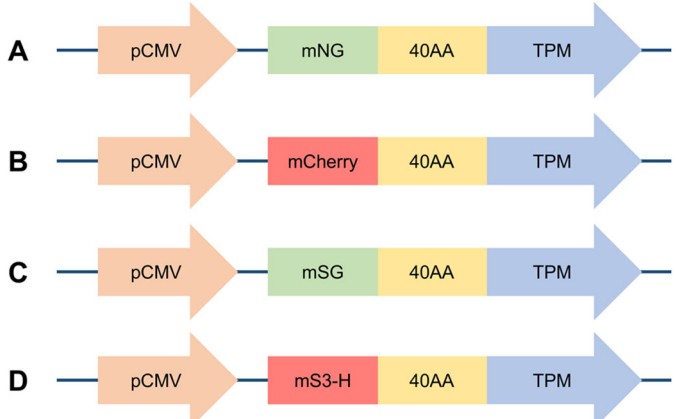

**Fig. 1. Tropomyosin fusion mammalian expression constructs used in this work.** Tropomyosin (TPM) N-terminally fused via a 40-amino acid flexible linker (40AA) to: (A) mNeonGreen (mNG); (B) mCherry; (C) mStayGold(E138D) (mSG); and (D) mScarlet3-H (mS3-H). For each, variants for nine different human tropomyosin isoforms were generated, namely TPM1.6, TPM1.7, TPM1.8, TPM1.12, TPM2.1, TPM2.2, TPM3.1, TPM3.2 and TPM4.2. All constructs are under the control of a CMV promoter (pCMV) for expression in mammalian cells.

colocalisation between mNG and phalloidin. Many have r values close to 1, signifying near identical fluorescence. TPM3.1 showed the lowest mean colocalisation, and TPM4.2 the highest. There are differences in the densities of tropomyosin and phalloidin fluorescence at some structures, particularly focal adhesions situated at the end of stress fibres, as exemplified in images of TPM1.6 and TPM3.1 (Fig. 2A), supporting recent similar findings (Kumari et al., 2024). The variation of this between images suggests isoform-specific differences in localisation density. Some images show small periodic gaps along stress fibres where phalloidin fluorescence is present but mNG significantly reduced, for example in cells expressing TPM1.7, while others show total co-localisation with actin, for example in TPM1.8 images. Striations like this have been observed in other fluorescent tropomyosin fusions (Sao et al., 2019; Meiring et al., 2019; Cagigas et al., 2025), but the basis of this localisation in unclear. The gaps in the striated patterning are ~1.5 µm (Fig. 2D). Length varies by isoform, but measurement of one high molecular weight isoform (PDB: 7UTL) (Pavadai et al., 2020) gives a length of 36.3 nm, suggesting 41 tropomyosin dimers could fit within 1.5 µm.

### Direct isoform comparison using different fluorescent proteins

For study of isoform-specific activity, different tropomyosin isoforms can be co-expressed and compared in a single cell, requiring different wavelength-excitable tags to distinguish isoforms. Versions of the 40AA-TPM construct with mCherry, the most widely used red fluorescent protein, were prepared for all nine isoforms (Fig. 1B) and imaged in live cells (Fig. 3). As expected, they highlighted cellular actin structures similarly to mNG. mCherry constructs were co-expressed with mNG constructs and imaged (Fig. 4), allowing direct comparison of several isoform pairs. Most isoforms highlighted the same structures, but with varying densities. Colocalisation of mCherry and mNG fluorescence demonstrates that presence of one isoform does not preclude another from assembling on the same actin bundle. Notable observed isoform differences include stronger TPM1.8 fluorescence in peripheral regions, while TPM2.1 fluorescence is stronger in the central cellular regions. TPM1.12 consistently produces less densely labelled actin structures and more diffuse

## Table 1. Human tropomyosin isoforms used in this work

| Isoform | Type | Gene | MW | NCBI Ref No. |
|---|---|---|---|---|
| TPM1.6 | α | TPM1 | High | NP_001018004.1 |
| TPM1.7 | α | TPM1 | High | NP_001018006.1 |
| TPM1.8 | α | TPM1 | Low | NP_001288218.1 |
| TPM1.12 | α | TPM1 | Low | NP_001018008.1 |
| TPM2.1 | β | TPM2 | High | NP_998839.1 |
| TPM2.2 | β | TPM2 | High | NP_003280.2 |
| TPM3.1 | γ | TPM3 | Low | NP_705935.1 |
| TPM3.2 | γ | TPM3 | Low | NP_001036816.1 |
| TPM4.2 | δ | TPM4 | Low | NP_003281.1 |

Type refers to the tropomyosin classification of each tropomyosin and MW refers to molecular weight classification of each tropomyosin.

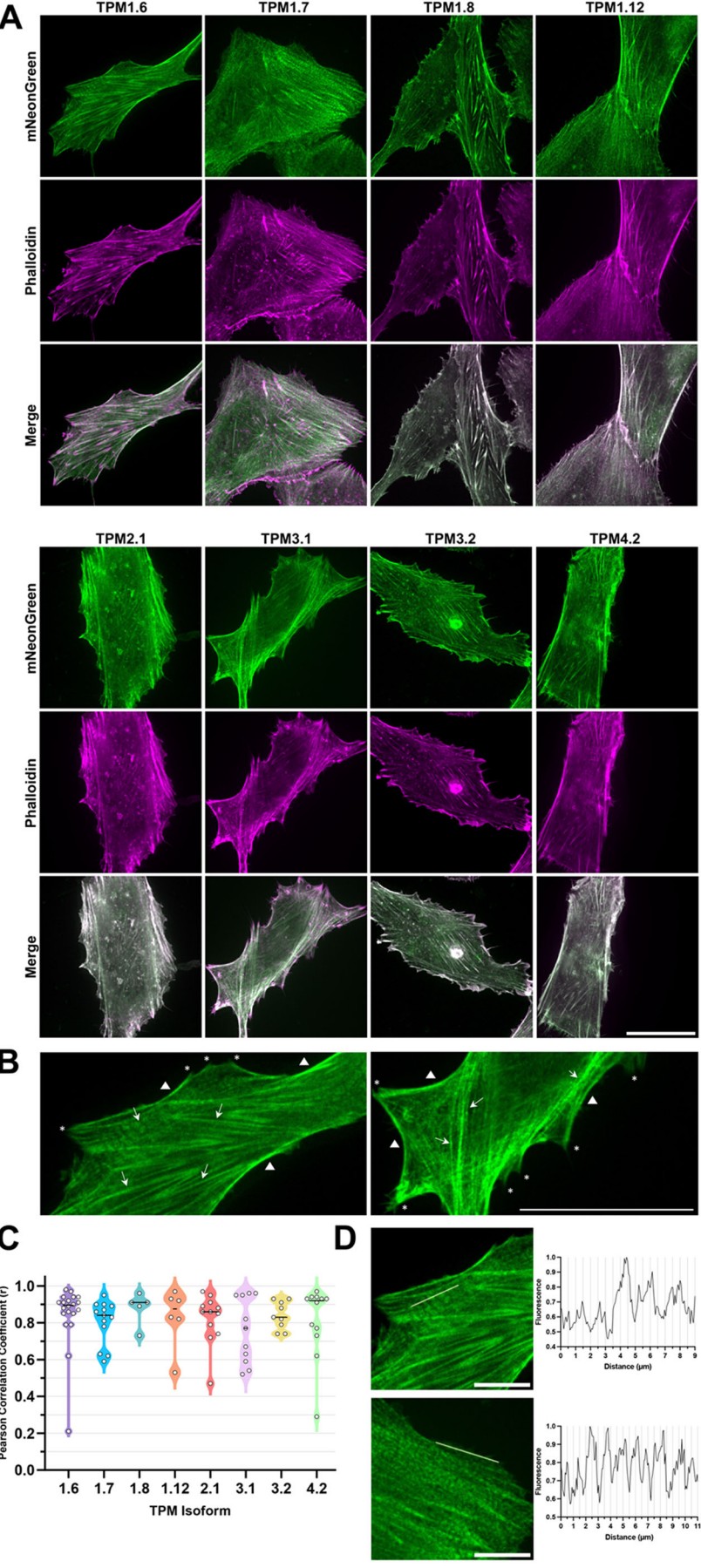

**Fig. 2. mNeonGreen-40AA-tagged tropomyosin isoforms colocalise with actin.** (A) Representative spinning-disk microscopy images of human tropomyosin isoforms fused via a 40-amino acid linker to mNeonGreen in hTERT-RPE1 cells. Note that the images shown are after fixation and staining with rhodamine-phalloidin. Scale bar: 30 µm (*n*=8). (B) Annotated images of cells expressing mNeonGreen-TPM1.6 (left) and mNeonGreen-TPM3.1 (right). Arrows point to stress fibres, triangles are adjacent to cortical actin, and asterisks indicate likely focal adhesions. Scale bar: 30 µm. (C) Strong colocalisation between mNG-TPM probes and phalloidin was detected in all isoforms using Pearson's correlation coefficient analysis. Each datapoint represents an image. (D) Fluorescence intensity plots of lines (shown in images) along stress fibres from cells expressing mNeonGreen-TPM1.6 (left) and mNeonGreen-TPM1.7 (right). Scale bars: 10 µm.

cytoplasmic fluorescence than other isoforms, this may be because TPM1.12 is a brain-specific isoform and therefore unable to incorporate effectively in an epithelial cell (Brettle et al., 2016; Lees-Miller et al., 1990), although the ordinary tropomyosin isoform expression profile of hTERT-RPE1 cells is yet to be characterised. TPM2.2 fluorescence was brightest on peripheral actin structures, and TPM4.2 is most densely localised at central stress fibres. Some images show an alternating pattern of isoforms on stress fibres, for example in mNG-TPM4.2/mCherry-TPM3.1 images. This phenomenon appears related to the earlier observed striations (Fig. 2) and has been previously seen in other non-muscle tropomyosin fusions (Sao et al., 2019; Meiring et al., 2019; Cagigas et al., 2025). Statistical approaches to isoform colocalisation studies can be combined with these tools. Pearson's correlation coefficient analysis was carried out on the images shown in Fig. 4, with two isoform pairs showing joint highest colocalisation, namely mNG-TPM2.2 and mCherry-TPM2.1, and mNG-TPM-1.8 and mCherry-TPM1.7. All images show a positive correlation between the two probes, but the lowest was seen in the mNG-TPM1.6 and mCherry-TPM1.12 image. Time-lapse imaging of mNG-TPM1.6/mCherry-TPM1.7-expressing cells (Fig. 5, Movie 1) shows normal cytoskeletal activity and persistent labelling of known tropomyosin-associated structures. Cell crawling can clearly be observed with lamellipodia forming on the leading edge, new stress fibres appearing in the new protrusions, and contacts with a neighbouring cell breaking as the cell is pulled forward. There are changes in probe expression as time post-transfection progresses, with mCherry becoming more prominent as chromophore maturation proceeds.

To test applicability of the tool, as well as to check whether the observed stress fibre striations are artefacts of the spinning-disk

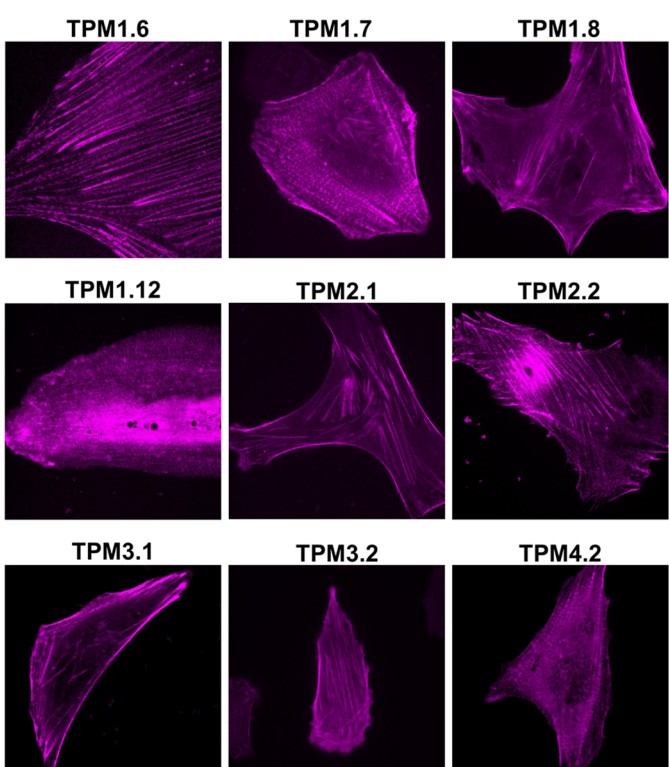

**Fig. 3. mCherry-40AA-tagged tropomyosin isoforms show typical tropomyosin fluorescence.** Representative spinning-disk microscopy images of human tropomyosin isoforms fused to mCherry via a 40-amino acid linker, expressed in live hTERT-RPE1 cells. Scale bar: 30 µm (*n*=8).

imaging modality, mNG and mCherry fusions were imaged via AiryScan super-resolution microscopy, which collects normally discarded light to improve resolution from 200 nm in regular confocal microscopy to 100 nm. AiryScan microscopy was used to image cells expressing both an mNG and an mCherry tagged tropomyosin (Fig. 6A), enabling acquisition of more defined images. Structured illumination microscopy (SIM) was employed to further improve resolution. SIM uses many angles of striped illumination to capture images up to 60 nm in resolution (Fig. 6B). The combined use of AiryScan and SIM imaging of mNG and mCherry-tagged tropomyosin confirms many of the findings observed via spinning-disk microscopy, including the stress fibre striations, which can be clearly seen in all combined images. A comprehensive study using these tropomyosin tools and super-resolution microscopy should deliver important information on organisational principles behind tropomyosin localisation and sorting.

### Fusion of human tropomyosin isoforms with photostable fluorescent proteins

StayGold is a highly photostable green fluorescent protein isolated from *Cytaeis uchidae* jellyfish (Hirano et al., 2022). Its photostability gives it applicability in time-lapse fluorescence microscopy, during which it resists photobleaching significantly more than mNG and sfGFP. StayGold is dimeric, which affects localisation of tagged proteins. We recently found a single residue change, E138D, that gives it monomeric behaviour [mStayGold(E138D)], which we successfully trialled using an mStayGold(E138D)-40AA-TPM2.2 construct (Ivorra-Molla et al., 2024). Here, we expanded this to the other eight tropomyosin isoforms (Fig. 1C), which were imaged via spinning-disk microscopy (Fig. 7A). The mStayGold(E138D) variants all showed strong actin structure staining, equivalent to that of mNG. These constructs are ideal for live cell time-lapse imaging and any fluorescence application that requires prolonged or significant laser exposure. In time-lapse imaging (Fig. 7B, Movie 2), the probes can be observed to show healthy cytoskeletal presentation and movement.

Finally, all nine of the selected TPM isoforms were fused to a recently developed red fluorescent mScarlet derivative with improved photostability named mScarlet3-H (Fig. 1D) (Xiong et al., 2025). These constructs were imaged in mammalian cells via spinning-disk microscopy (Fig. 8). Clear tropomyosin labelling was observed with mScarlet3-H. Less overall definition was seen than the previous probes due to lower cellular expression and brightness. Sequence engineering can optimise mammalian mScarlet3-H expression. Due to this protein's high photostability over other red fluorescent proteins, these mScarlet3-H-TPM constructs are a promising toolset for multicolour long-term image acquisition of human tropomyosin isoforms.

### DISCUSSION

In summary, adding to an original TPM2.2 construct, we tagged mNG to eight human tropomyosin isoforms via 40-amino acid linkers to reduce potential perturbation of protein activity. We generated mCherry, mStayGold(E138D) and mScarlet3-H versions of these nine constructs, and imaged all 36 constructs in live mammalian cells via spinning-disk microscopy, which showed actin cytoskeleton labelling typical of tropomyosin. mNG and mCherry-tagged isoforms were co-expressed in cells and imaged via spinning-disk microscopy, as well as super-resolution microscopy techniques, AiryScan and SIM. This produced high

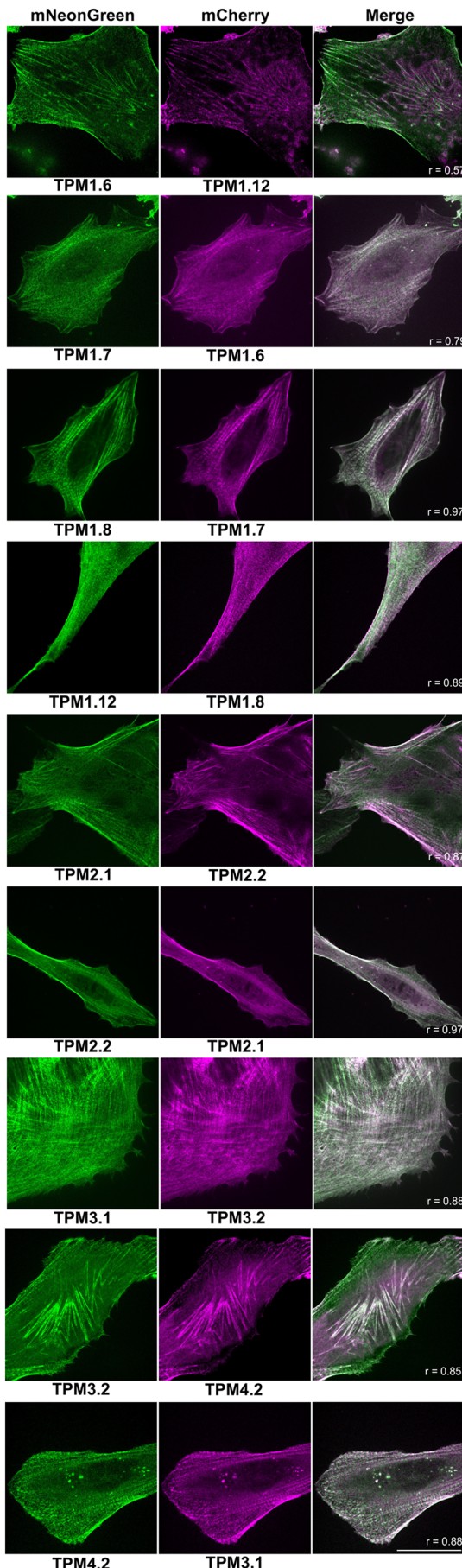

**Fig. 4. Tropomyosin isoforms tagged with mNeonGreen and mCherry via a 40-amino acid linker allow direct *in vivo* comparison of isoform localisation.** Representative spinning-disk microscopy images of live hTERT-RPE1 cells co-expressing two different human tropomyosin isoforms fused to mCherry and mNeonGreen via a 40-amino acid linker. Scale bar: 30 µm (*n*=11). Colocalisation for each image shown was quantified by Pearson's correlation coefficient (r) method.

resolution comparisons of isoform localisation within single cells and revealed isoform-specific differences in localisation density. Striations in fluorescence of tropomyosin observed along stress fibres require further biochemical work to uncover their structural basis, as does the persistently less dense fluorescence of actin structures labelled with TPM1.12, which may have lower binding affinity than other isoforms. Other differences seen in isoform comparisons also warrant further analysis, and swapping fluorescent tags between each isoform pair will test whether findings are due to expression or construct stability. These findings act as proof of principle for the applicability of this tool. The use of dual colour imaging of different isoforms simultaneously allows for investigations of whether different tropomyosins reside in the same actin bundle (for example, stress fibres contain 10-30 actin filaments).

Translating this improved tool from yeast to human cells has facilitated human isoform-specific tropomyosin comparison *in vivo*. It shows little sign of mislocalisation, and all three construct groups express strongly for easy acquisition of high-contrast images. The next step in expanding the tool further would be to generate metazoan lines stably expressing the constructs to enable study tissue-specific isoform functions in a multicellular organism. As a terminal tag fused at sites important for tropomyosin polymerisation, there may be perturbation that is not easily observable. The ideal tropomyosin tag would be mid-chain, in a position of no importance for tropomyosin structure, function and partner interactions. The difficulty of mid-sequence tags is the necessity to laboriously screen different sites and identify an optimal tag position. There are a number of mid-chain tags available: tetracysteine tags which can be bound by fluorescent arsenic-based reagents such as ReAsH and FlAsH (Luedtke et al., 2007); unnatural amino acids which undergo click reactions with clickable dyes, for example TCO*-Lysine with SiR-tetrazine dye (Kozma et al., 2016); and immuno-stainable tags, such as ALFA tag (Götzke et al., 2019). Optimal mid-chain tagging would remove blockage of protein–protein interactions, however perturbation of cell morphology by transient transfection may also affect observations, which could be addressed through stable transfection.

These constructs tackle the current lack of isoform-specific tools for imaging tropomyosin, with most antibodies not having specificity to a single isoform (Hatano et al., 2022). By targeting antibodies against the fluorescent protein, techniques that rely on immunostaining can be combined with this tag, including further super-resolution techniques such as dSTORM and expansion microscopy (Klein et al., 2011; Chen et al., 2015). Another potential application is isoform-specific purification of tagged tropomyosins using antibodies or beads against the fluorescent proteins. This work provides evidence for the use and underutilisation of lengthy polypeptide linkers as a tool for optimal terminal tagging of target proteins with fluorescent markers. Flexible linkers should be considered for fusions of any polymeric protein, such as actin, tubulin, keratin and septin. Linker design can be aided by machine learning approaches, particularly when looking to ensure undisturbed protein-protein interactions (Xu et al., 2024).

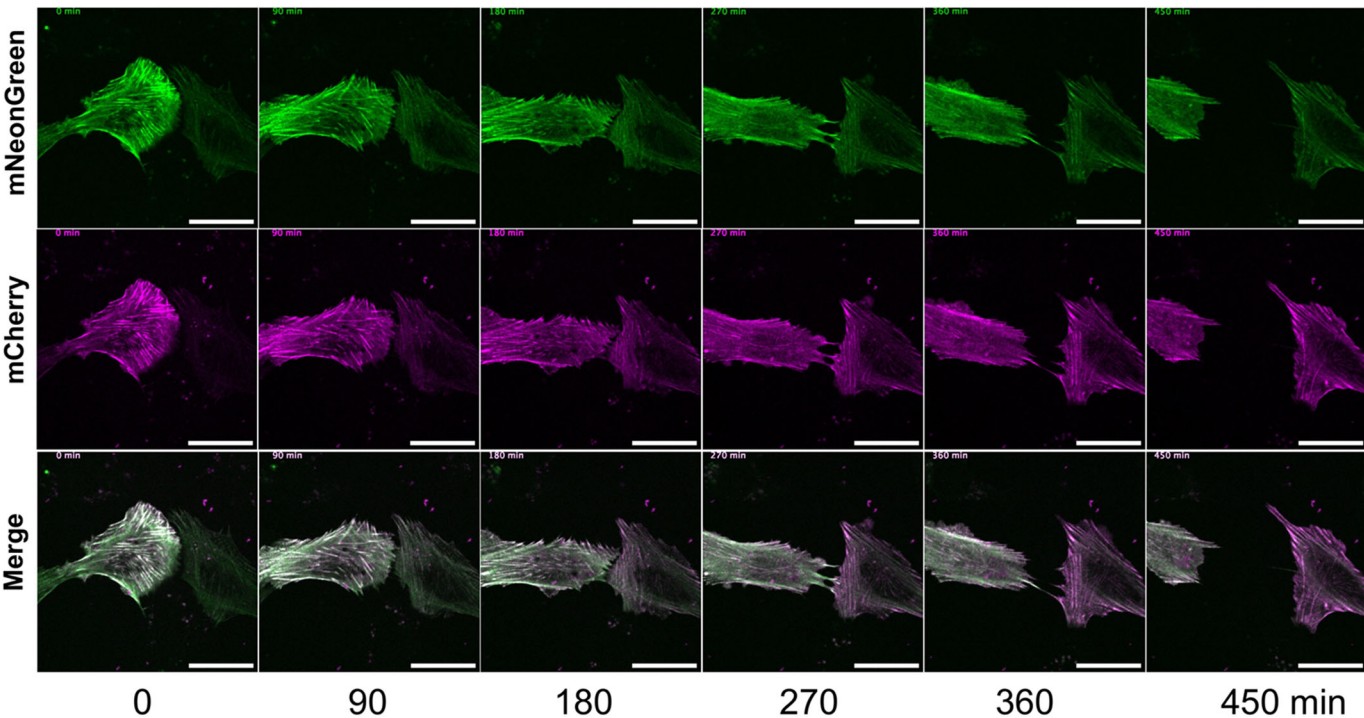

**Fig. 5. Individual frames from time-lapse imaging of cells expressing mNG-TPM1.6 and mCherry-TPM1.7 show normal cytoskeletal activity.** Images were taken every 5 min; scale bars: 30 μm (*n*=10).

## MATERIALS AND METHODS

### Plasmid construction

To generate mNeonGreen-40AALinker-TPM constructs, pmNeonGreenHO-G (AddGene, #127912) vector was linearised via restriction digestion with BspEI (New England Biolabs, #R0540). Human-codon-optimised G-blocks (IDT) were synthesised for each of the chosen tropomyosin isoforms (TPM 1.6, 1.7, 1.8, 1.12, 2.1, 2.2, 3.1, 3.2 and 4.2) N-terminally fused to the 40-amino acid flexible linker (LEGSGQGPGSGQGSGSPGSGQGPG-SGQGSGPGQ GSGPGQG) with overhang sequences complementary to the linearised vector on either end. The G-blocks were each inserted into the linearised vector via Gibson assembly with NEBuilder® HiFi DNA Assembly (New England Biolabs, E2621) and transformed into chemically competent DH5α *E. coli*. Plasmids were purified using QIAprepSpin Miniprep kit (Qiagen, 27104) and screened via restriction digestion, agarose gel electrophoresis, Sanger sequencing, and Nanopore whole-plasmid sequencing. mCherry, mStayGold(E138D) and mScarlet3-H versions of these constructs were generated by PCR-linearisation of each mNeonGreen-40AALinker-TPM construct, removing mNeonGreen and into its place inserting a synthetic mCherry, mStayGold(E138D) or mScarlet3-H G-block with complementary overhangs. These were transformed and screened as before.

### Cell culture

Immortalised diploid human retinal pigment epithelial (hTERT-RPE1) cells (ATCC; CRL-4000) were cultured in DMEM/Nutrient Mixture F-12 Ham supplemented with 15 mM HEPES, sodium bicarbonate (Sigma-Aldrich, D6421), 10% FBS (Sigma-Aldrich, F7524), 0.365 g/l L-glutamine (Gibco, 25030024), and 100 U/ml penicillin/streptomycin (Gibco, 15140122) at 37°C with 5% $CO_2$. LookOut Mycoplasma PCR Detection Kit (Sigma-Aldrich, MP0035) with JumpStart Taq Polymerase (Sigma-Aldrich, D9307) was used for regular mycoplasma testing via agarose gel electrophoresis.

### Transfection

Wells of $3.1×10^4$ hTERT-RPE1 cells were seeded in 200 μl media each on eight-well chamber μ-slides (IBIDI, 80826) 18-24 h prior to transient transfection. For transfection of each wall, 0.5 μg total TPM plasmid (0.25 μg each if two plasmids are used) was complexed with 1 μl Lipofectamine 2000 (Invitrogen, 11668019) in 75 μl OptiMem media (Gibco, 31985070) for 20 min, before addition to cells. After 6 h, media was replaced with regular growth media, and image acquisition was carried out after a further 18 h.

### Fixation and phalloidin staining

For phalloidin-stained images, 24 h post-transfection, hTERT-RPE1 cells were fixed in 4% paraformaldehyde in PBS for 30 min, followed by three PBS washes. Using 0.1% Triton X-100 in PBS, rhodamine-conjugated phalloidin (Invitrogen, R415) was diluted 1:400 and added to cells for 90 min. Finally, the samples were washed with PBS and sealed with Vectashield® (Vector, H-1000) and imaged.

### Spinning-disk confocal microscopy

All images other than the super-resolution images in Fig. 6, were acquired using one of the two following spinning-disk confocal microscopy set ups: (A) An Andor Revolution XD system equipped with a Nikon ECLIPSE Ti inverted microscope, a CSU-X1 Yokogawa spinning-disk system, an Andor iXon Ultra EMCCD camera, a Nikon Plan Fluor 40×1.30 NA oil-immersion objective lens (200 nm/pixel), and Andor IQ3 software. (B) An Andor TuCam system, equipped with a Nikon ECLIPSE Ti inverted microscope, a CSU-X1 Yokogawa spinning-disk system, two Andor iXon Ultra EMCCD cameras, a Nikon Plan Apo Lambda 100×1.45 NA oil immersion objective lens (69 nm/pixel), and Andor IQ3 software. In both systems, mNeonGreen and mStayGold(E138D) were excited by a 488 nm wavelength laser line, while mCherry and mScarlet3-H were excited by a 561 nm wavelength laser line. Image acquisitions were carried out at 37°C for live cells. The images shown are a mixture of maximum intensity Z-projections and single slices. All images in this paper were processed using ImageJ software. For Pearson's correlation coefficient analysis, ImageJ's built-in 'coloc 2' feature was used to compare colocalisation of two channels in an image.

### Time-lapse imaging

hTERT-RPE1 cells were seeded on fibronectin-coated eight-well chamber μ-slides and transfected as described above. Six h post-transfection the cell media was replaced with PhenolRed-free Leibovitz's L-15 Medium (Gibco,

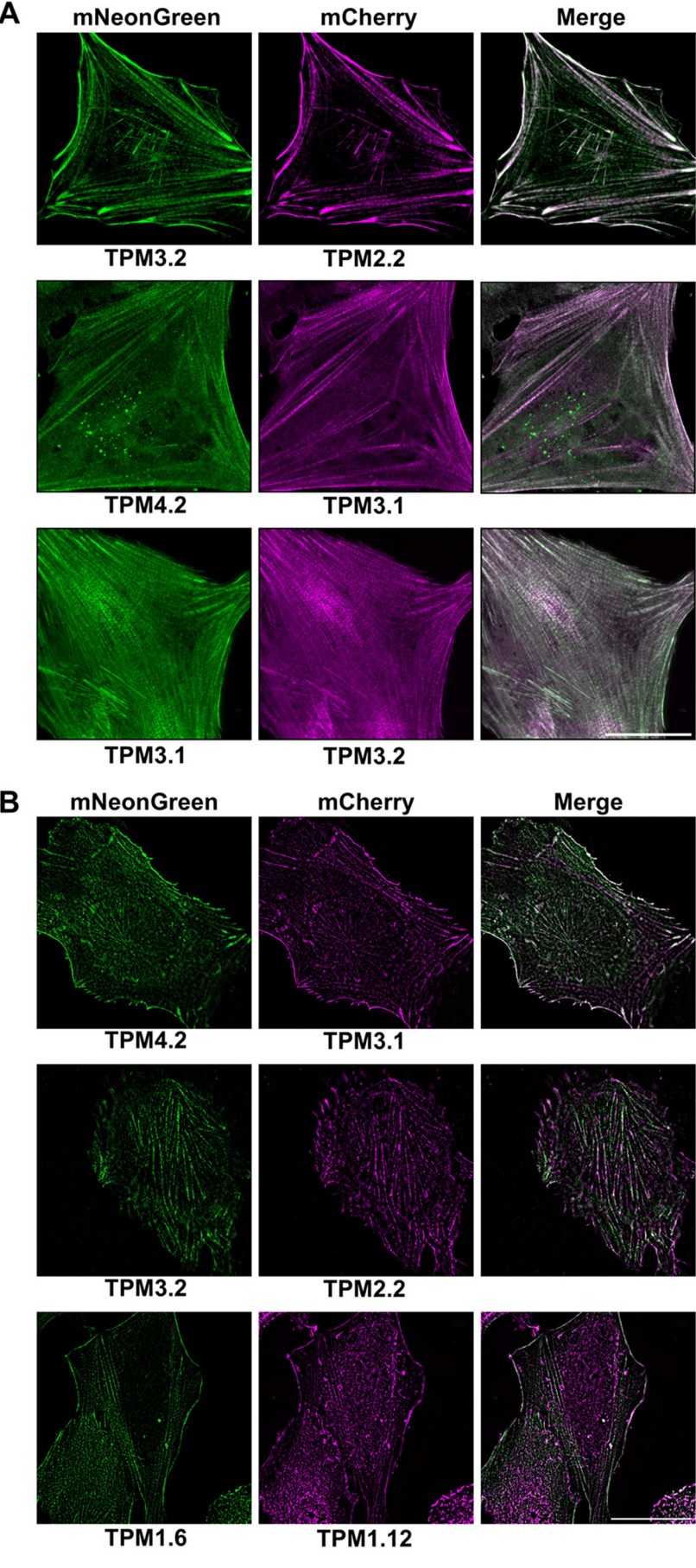

**Fig. 6. Tagging with flexible 40-amino acid linkers facilitates direct super-resolution comparison of human tropomyosin isoforms.** Representative super-resolution microscopy of fixed hTERT-RPE1 cells co-expressing tropomyosin isoforms fused via a 40-amino acid linker to mCherry and mNeonGreen. (A) AiryScan microscopy images. (B) Structured illumination microscopy images. Scale bars: 30 µm (*n*=5).

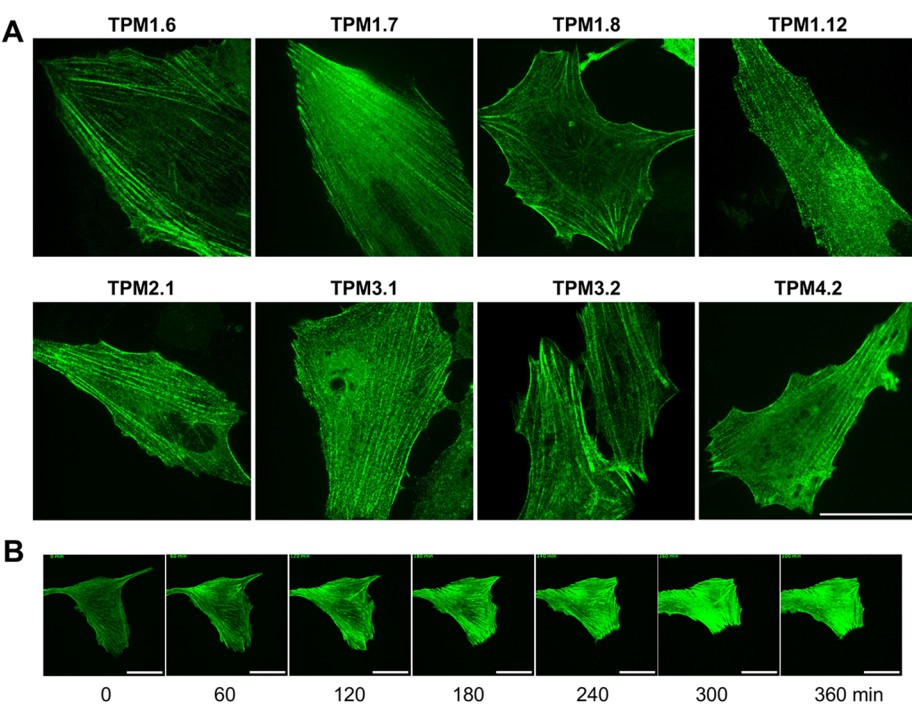

**Fig. 7. Tropomyosins tagged with a photostable green fluorescent protein via a 40-amino acid linker are a new tool for applications requiring prolonged laser exposure.** (A) Representative spinning-disk microscopy images of live hTERT-RPE1 cells expressing tropomyosin isoforms fused via a 40-amino acid linker to mStayGold(E138D) (*n*=9). (B) Individual frames from time-lapse imaging of cells expressing mStayGold(E138D)-TPM1.7. Images were taken every 5 min (*n*=10). Scale bars: 30 μm.

21083027) and the cells were immediately imaged under the above spinning-disk microscopy setup at 37°C for 24 h, with acquisition every 5 min.

### Structured illumination microscopy (SIM)
Images were acquired on a Zeiss Elyra 7 system, using SIM grating periods of 36.5 μm with 13 modulations. A 488 nm laser with 3% attenuation was used for mNeonGreen excitation and a 561 nm laser with 4.5% attenuation

was used for mCherry excitation. Lasers were aligned using slide-mounted glass beads. A 570-620 nm emission filter was used with the 488 nm laser and a 495-550 nm emission filter was used with the 561 nm laser. A 63×1.46 NA oil-immersion Plan-Apochromat objective was used for this. Data processing was carried out with the proprietary algorithm of Zeiss Zen. The shown images are a mixture of maximum intensity Z-projections and single slices.

### AiryScan microscopy
Images were acquired on a Zeiss LSM980, using the AiryScan detector. A 488 nm laser with 0.4% attenuation was used for mNeonGreen excitation and a 561 nm laser with 0.4% attenuation was used for mCherry excitation. A 63×1.46 NA oil-immersion Plan-Apochromat objective was used for this. Data processing was carried out with the proprietary AiryScan processing algorithms of Zeiss Zen. The shown images are a mixture of maximum intensity Z-projections and single slices.

### Acknowledgements
The authors thank staff from the BioSLRs in the School of Life Sciences at the University of Warwick for use of the Zeiss Elyra 7 microscope funded by BBSRC Alert and the Zeiss LSM980 AiryScan microscope. We also thank staff at the Computing and Advanced Microscopy Unit (CAMDU) at the University of Warwick for maintenance of the spinning-disk microscopes used in this work. Furthermore, we thank Teresa Massam-Wu for valuable advice.

### Competing interests
The authors declare no competing or financial interests.

### Author contributions
Conceptualization: W.S., M.K.B.; Formal analysis: W.S., V.P.; Funding acquisition: M.K.B.; Investigation: W.S., V.P., J.M., I.H.-P., M.K.B.; Supervision: M.K.B.; Validation: W.S.; Writing – original draft: W.S., M.K.B.; Writing – review & editing: I.H.-P., M.K.B.

### Funding
W.S. was funded by the MRC (MR/N014294/1), the Wellcome Trust (311433/Z/24/Z), the Medical and Life Sciences Research Fund, and a University of Warwick Institute of Advanced Study Early Career Fellowship. V.P. was funded by the Human Frontier Science Program (RGP001/2023) awarded to M.K.B. via the *Scientists4Scientists* initiative. I.H.P. was funded by the BBSRC. M.K.B. was funded by a Wellcome Trust Senior Investigator Award (WT101885MA) and Bioimaging grant (311433/Z/24/Z). Open Access funding provided by University of Warwick. Deposited in PMC for immediate release.

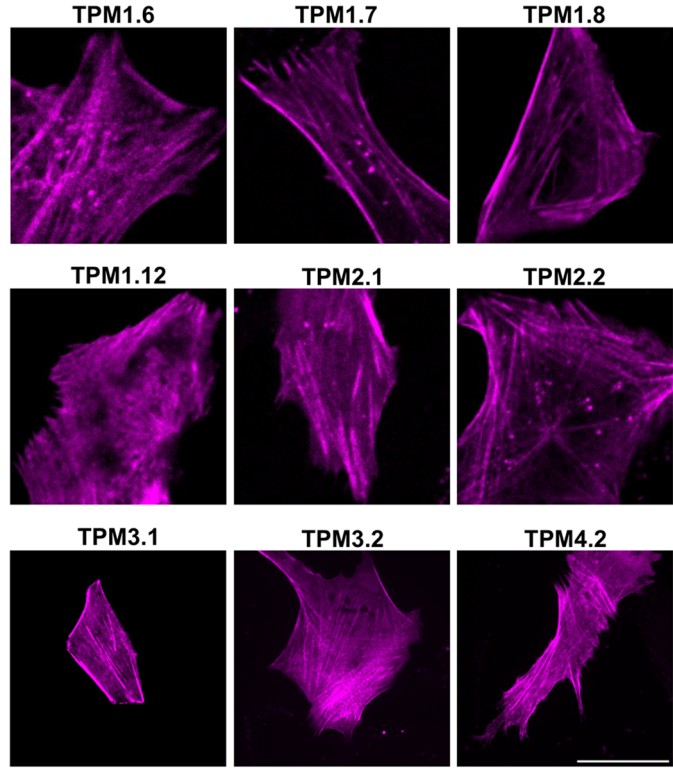

**Fig. 8. Tagging of an improved photostability red fluorescent protein with tropomyosins via a 40-amino acid linker.** Representative spinning-disk microscopy images of live hTERT-RPE1 cells expressing different tropomyosin isoforms fused to mScarletH-3 via a 40-amino acid linker (*n*=5).

## Data and resource availability
All tropomyosin constructs and sequences used in this work are available on request.

## Peer review history
The peer review history is available online at https://journals.biologists.com/bio/lookup/doi/10.1242/bio.061992.reviewer-comments.pdf

## First Person
This article has an associated First Person interview with the first author of the paper.

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
