## [Peer Review File · Biology Open]

Fluorescent protein tags for human tropomyosin isoform comparison

Will Scott, Vitaliia Polutranko, Jakub Milczarek, Ian Hands-Portman and Mohan K.

Balasubramanian

DOI: 10.1242/bio.061992

Editor: Catherine L. Jackson

Review timeline

Original submission: 19 March 2025

Editorial decision: 16 April 2025

First revision received: 26 June 2025

Accepted: 2 July 2025

Original submission

First decision letter

MS ID#: bio.061992

MS TITLE: Fluorescent protein tags for human tropomyosin isoform comparison

AUTHORS: Will Scott, Vitaliia Polutranko, Jakub Milczarek, Ian Hands-Portman and Mohan K. Balasubramanian

I have now reached a decision on the above manuscript.

The reviewer reports are shown at the bottom of this email or can be accessed, together with a copy of this decision letter, by going to:

As you will see, the reviewers gave favourable reports, but raised some critical points that will require amendments to your manuscript. I hope that you will be able to carry these out, because we would like to be able to accept your paper.

At this stage, we also ask you to ensure your manuscript complies with our formatting guidelines "please see our manuscript preparation guidelines for details. Provided you are able to fully address the referees' comments, we are positive about publication of your paper (we accept over 95% of revision submissions) and therefore hope you won't mind any extra work involved in reformatting your manuscript at this point.

Please ensure that you clearly highlight all changes made in the revised manuscript. Please avoid using 'Tracked changes' in Word files as these are lost in PDF conversion.

I should be grateful if you would also provide a point-by-point response detailing how you have dealt with the points raised by the reviewers in the 'Response to Reviewers' box. Please attend to all of the reviewers' comments. If you do not agree with any of their criticisms or suggestions please explain clearly why this is so.

Reviewer 1

Comments for the author

This manuscript addresses the feasibility of generating fluorescently tagged derivatives of tropomyosin isoforms that can be used to report on the intracellular location of these isoforms. The value of this research is high because the tropomyosin isoforms determine the functional capacity of their associated actin filaments. The authors report on the construction of N-terminal tagged tropomyosins with 3 different reporters and use a 40 amino acid linker to minimise the impact of the tag. Extensive imaging is presented which demonstrates the association of the tagged proteins with actin filaments in cultured cells. This library of constructs is likely to be widely used by the research community. There are several issues which should be considered by the authors.

1. On page 3 it is stated that 'larger tropomyosins are known to bind actin more stably than shorter isoforms. Of the cytoskeletal isoforms the greatest avidity for actin is found for Tpm1.8 which is a shorter isoform (Janco et al, BioArchitecture, 2016). It might be better to avoid any declarative statement.
2. Page 4 is the first mention of the hTERT-RPE1 cell line. It would help the reader to explain what these cells are; retinal pigment epithelial cells.
3. The cells used for transfection experiments are human retinal pigment epithelial cells. Are the authors aware of the tropomyosin isoforms normally expressed by these cells. For example, Tpm1.12 is brain specific and would not be expected to be expressed in these retinal epithelial cells. The test could be made a little more explicit that in the case of Tpm1.12, the question is whether it incorporates into filaments in an epithelial cell.
4. The observation of TPM1.6 and 3.1 (Fig 2A, end page 4, beginning page 5) associated with focal adhesions should reference the recent paper from the Lappalainen lab (Kumari et al, Nat Comms, 2024).
5. In paragraph 1, page 5, there is mention of striations. Despite my conflict of interest (COI), I think it may be relevant to mention Meiring et al, J Cell Sci, 2019 who demonstrate the striated organisation of Tpm3.1/2 using electron microscopy. However, due to COI, feel free to ignore.
6. Despite my COI, the recent observation of alternating Tpm3.1/Tpm4.2 in stress fibres has recently been reported using the respective antibodies in mouse embryonic fibroblasts which provides strong support for the authors' observations (Cagigas et al, Cytoskeleton, 2025). However, due to COI, feel free to ignore.

Reviewer 2

Comments for the author

In this study from the Balasuramanian lab that initially discovered the use of a 40 AA linker that permits fluorescent protein tagging of tropomyosins. They subsequently used this tag with monomeric StayGold and here they have used this same tag to study other Tpm isoforms and with different fluorescent proteins. This is a logical if not large leap in progress, but still represents a valuable piece of work for the Tpm and cell biology community. The manuscript is well written and brief, which means that it lacks a few controls and deeper insights.

Analysis missing:

Could the authors present a statistical approach to colocalization? Maybe the Manders or Pearson tests?

How far apart are the striations on the Tpm's, can we see a good analysis?

Can the discussion go a little further into the potential role of the dual labeling of the same stress fibers?

What are the statistics in the imaging, we only see single cells, there is no indication of the cell by cell variation.

Reviewer's Responses to Questions

Experimental quality

Does each figure have the proper controls?

If 'No', please indicate reasons in Comments for Author box below.

Reviewer #1:

No

Reviewer #2:

Yes

Were the data analyzed using appropriate statistical tests?
If 'No', please indicate reasons in Comments for Author box below.

Reviewer #1:

Yes

Reviewer #2:

Yes

Reproducibility

Were experiments performed using adequate number of biological replicates?
If 'No', please indicate reasons in Comments for Author box below.

Reviewer #1:

Yes

Reviewer #2:

No

Does the methods section provide sufficient detail to permit reproducibility?
If 'No', please indicate reasons in Comments for Author box below.

Reviewer #1:

No

Reviewer #2:

No

Completeness

Are the manuscript's conclusions supported by the data?
If 'No', please indicate reasons in Comments for Author box below.

Reviewer #1:

No

Reviewer #2:

Yes

Scholarship

Do the authors cite and discuss the merits of data that would argue for and against their conclusion?
If 'No', please indicate reasons in Comments for Author box below.

Reviewer #1:

Yes

Reviewer #2:

No

First revision

Author response to reviewers' comments

Dear Dr Jackson,

Thank you for providing the reviewer's comments on our manuscript. We agree with all comments and have made all recommended changes, as detailed below (our responses in bold):

Reviewer 1:

"There are several issues which should be considered by the authors.

1. On page 3 it is stated that 'larger tropomyosins are known to bind actin more stably than shorter isoforms. Of the cytoskeletal isoforms the greatest avidity for actin is found for Tpm1.8 which is a shorter isoform (Janco et al, BioArchitecture, 2016). It might be better to avoid any declarative statement."

It has now been clarified that there are exceptions to our statement, such as TPM1.8. (lines 20-23 page 3).

"2. Page 4 is the first mention of the hTERT-RPE1 cell line. It would help the reader to explain what these cells are; retinal pigment epithelial cells."

An explanation of the cell type has now been included (line 31 page 4).

"3. The cells used for transfection experiments are human retinal pigment epithelial cells. Are the authors aware of the tropomyosin isoforms normally expressed by these cells. For example, Tpm1.12 is brain specific and would not be expected to be expressed in these retinal epithelial cells. The test could be made a little more explicit that in the case of Tpm1.12, the question is whether it incorporates into filaments in an epithelial cell."

We have now mentioned this and linked it to potential expression issues with new references (lines 31-34 page 5).

"4. The observation of TPM1.6 and 3.1 (Fig 2A, end page 4, beginning page 5) associated with focal adhesions should reference the recent paper from the Lappalainen lab (Kumari et al, Nat Comms, 2024)."

This reference has now been cited (line 7 page 5).

"5. In paragraph 1, page 5, there is mention of striations. Despite my conflict of interest (COI), I think it may be relevant to mention Meiring et al, J Cell Sci, 2019 who demonstrate the striated organisation of Tpm3.1/2 using electron microscopy. However, due to COI, feel free to ignore."

This reference has now been cited (line 12 page 5).

"6. Despite my COI, the recent observation of alternating Tpm3.1/Tpm4.2 in stress fibres has recently been reported using the respective antibodies in mouse embryonic fibroblasts which provides strong support for the authors' observations (Cagigas et al, Cytoskeleton, 2025). However, due to COI, feel free to ignore."

This reference has now been cited (line 13 page 5).

Reviewer 2:

“Could the authors present a statistical approach to colocalization? Maybe the Manders or Pearson tests?”

Pearson tests have now been carried out for all colocalisation experiments in the manuscript, namely between mNG-TPM and phalloidin-rhodamine in Fig. 2C, and between mNG-TPM and mCherry-TPM in Fig. 4. (lines 1-4 page 5; lines 5-10 page 6;)

“How far apart are the striations on the Tpm's, can we see a good analysis?”

This analysis is carried out in Fig. 2D and is ~1.5 μ m and described in paragraph 1 on page 5.

“Can the discussion go a little further into the potential role of the dual labeling of the same stress fibers?”.

Our reasoning for dual labelling is to determine whether different TPM isoforms can reside in the same stress fiber. Note that stress fibers can have 10-30 actin filaments in a bundle and dual color labelling can help determine how isoforms distribute on a single bundle (lines 3-5 page 8).

“What are the statistics in the imaging, we only see single cells, there is no indication of the cell by cell variation.”

The number of cells screened in each experiment has now been added to all figure legends.

Please note that we have included an additional Figure, with a fourth set of constructs featuring TPM isoforms fused to mScarlet3-H, a red fluorescent protein with improved photostability. This fluorescent protein tagged tropomyosins became possible since between submission of our paper and now a new paper appeared describing this photostable RFP, termed mScarlet3-H. We are happy to remove it from the paper if the editor advises so, since this part wasn't peer reviewed. A new author Jakub Milczarek, who made all the Scarlet constructs has been added.

We look forward to hearing from you and hope the revisions will satisfy you and the referees. We are looking forward to seeing our manuscript published soon.

Second decision letter

MS ID#: bio.061992

MS TITLE: Fluorescent protein tags for human tropomyosin isoform comparison

AUTHORS: Will Scott, Vitaliia Polutranko, Jakub Milczarek, Ian Hands-Portman and Mohan K. Balasubramanian

I am happy to tell you that your manuscript has been accepted for publication in Biology Open, pending our standard publication integrity checks. It was accepted on 02 July 2025.